# Seasonality and Phosphate Fertilization in Carbohydrates Storage: *Carapa guianensis* Aubl. Seedlings Responses

**DOI:** 10.3390/plants11151956

**Published:** 2022-07-27

**Authors:** Vanessa Leão Peleja, Poliana Leão Peleja, Túlio Silva Lara, Edgard Siza Tribuzy, José Mauro Sousa de Moura

**Affiliations:** 1Institute of Biodiversity and Forests, Federal University of Western Pará, Santarém CEP 68040-255, Brazil; polianalepeleja5@gmail.com (P.L.P.); estribuzy@gmail.com (E.S.T.); 2Institute of Water Science and Technology, Federal University of Western Pará, Santarém CEP 68040-255, Brazil; tulio.lara@yahoo.com.br; 3Interdisciplinary Training Center, Federal University of Western Pará, Santarém CEP 68040-255, Brazil; jmaurosm@gmail.com

**Keywords:** water availability, phosphor, plant biomass, carbohydrate allocation, *C. guianensis*

## Abstract

The low availability of phosphorus and water in soil can promote the remobilization of carbohydrates in the plant, releasing energy to mitigate stress. In this context, our objective was to analyze the production and allocation of carbohydrates in plants of *Carapa guianensis* Aubl. submitted to different doses of phosphate fertilization, during the rainy and dry seasons, in the western region of Pará. We used three phosphorus dosages (0, 50, 250 kg ha^−1^) as treatments. We evaluated the plants during the dry and wet seasons. We quantified dry matter production, phosphorus content, total soluble sugars, reducing sugars, sucrose, and starch. Phosphate fertilization and different evaluation periods influenced carbohydrate concentrations (*p* < 0.05) in plants. The highest levels of P in the leaves were registered in October and, in the roots the content decreased with the passage of time in all treatments. The control had higher dry matter production in leaves and stems. During the dry season, there was an accumulation of carbohydrates in plants and a low production of dry matter. Soluble sugars and sucrose tended to be allocated to the stem, reducing sugars to the leaves and starch to the roots, in most periods. In general, *C. guianensis* seedlings were not very responsive to phosphorus addition.

## 1. Introduction

During the life cycle, plants adapt to the environment they inhabit, mainly due to the availability of nutrients and water, considerably influencing plant growth. Environmental factors result in the development of strategies for survival, such as rapid growth to increase efficiency in carbohydrate metabolism [1,2,3].

Carbohydrates perform several functions in the plant, acting as energy transport molecules, signaling abiotic stresses (similar to hormones), osmotic adjustments, and growth, among others [4]. The carbohydrates allocation in the plant is driven by nutritional requirements and controlled by biogeochemical constraints (air, temperature, water, etc.). However, storage and remobilization are influenced by nutritional and water factors that still need to be studied. 

Among the macronutrients, phosphorus (P) is an essential nutrient and directly influences the development of plants. In acidic, weathered soils rich in Al and Fe oxides, predominant in tropical regions, inorganic phosphorus (Pi) is strongly adsorbed on the edges of silicate clay minerals and oxides, which makes the nutrient unavailable to plants [5]. When this nutrient is limited, it impairs photosynthesis due to the low availability of Pi in the chloroplast, which leads to reduced carbon assimilation and results in lower production of triose phosphate, used in carbohydrate synthesis [2,6]. 

Plant growth, P dynamics, and carbohydrate distribution vary between species and are directly influenced by climatic conditions, especially precipitation. Experiments with forest species show different patterns of growth, storage, and distribution of carbohydrates under the same seasonal conditions. Drought and low nutrient content can trigger the accumulation of soluble sugars (sucrose, glucose, and fructose) [7,8] or a decrease in reserves [9]. Thus, the characterization of carbohydrate distribution helps in understanding the performance and productivity of trees and seedlings, considering that the distribution between organs (leaf, stem, root) is vital for the plant′s adaptation to the environment [9]. 

The influence of phosphate fertilization and seasonality on the production and allocation of sugars in *Carapa guianensis* Aubl plants is still poorly understood. Species of the genus *Carapa* are valued for their high-quality wood and the oil extracted from their pharmaceutically owned seeds. They occur in tropical forests in Africa and South America but grow in a variety of forest types such as savannas and floodplains [10,11].

Considering the adaptability of the species to different environments and the functions of P and carbohydrates in plant survival, our objective was to analyze the production and allocation of carbohydrates in plants of *Carapa guianensis* Aubl. submitted to different doses of phosphate fertilization in different seasonal periods, in the Western Region of Pará. The guiding questions of this study were: (1) Does the addition of phosphorus to the soil stimulate the biomass production of *C. guianensis* seedlings? (2) Does phosphorus influence carbohydrates production in *C. guianensis* seedlings? (3) How do seasonal variations affect carbohydrate dynamics in *C. guianensis* seedlings?

## 2. Results

The interaction between phosphate fertilization levels and evaluation periods was significant (*p* < 0.01). There were influences on dry matter production, concentrations of total soluble sugars-AST (*p* < 0.001) and reducing sugars-AR (*p* < 0.001), and sucrose (*p* < 0.001), starch (*p* < 0.0016) from leaves, stem and roots.

### 2.1. Phosphorus Content

The highest levels of P in the leaves were recorded in October. In the roots, the element content decreased over time in all treatments. In the control, the foliar P content did not vary between the periods, except in December. In plants treated with 250 kg ha^−1^ of P, the content of the element in the leaves increased until the month of October, the dry season, and then decreased (Figure 1).

### 2.2. Dry Matter

The production of dry matter in leaves, stems, and roots varied between treatments according to each collection period, with a gradual increase over time. During the dry season, August and October, the plants showed low growth, followed by a significant increase above 100% in the last collection (February), rainy season, expressing itself mainly in the stem (Figure 2).

The distribution pattern of leaf and stem dry mass of the plants was similar. In the dry season, treatment plants with 250 kg ha^−1^ of P had the lowest averages of dry matter. In the last evaluation (February), the control and the treatment 50 kg ha^−1^ of P had the highest averages of leaf and stem dry mass. However, in the roots, the highest production occurred in fertilized plants.

### 2.3. Carbohydrate Distribution

#### 2.3.1. Total Soluble Sugars (TSS)

The concentration of total soluble sugars (TSS) in leaves, stems, and roots varied between treatments and according to each collection period, except for the stem—where no influence of phosphorus was observed on soluble sugar concentrations (*p* > 0.05). 

In October, during the dry season, there was a drop in foliar concentrations of TSS. On the other hand, in the stem and roots, there was an increase compared to the previous period. During this period, the highest soluble sugar reserves were recorded in the stem, however, the concentrations were not influenced by phosphorus (Figure 3).

Plant TSS partition in October averaged 25% in leaves, 46% in the stem, and 29% in the root, between treatments. In February, the rainy season, an average of 38% of the TSS was allocated to the leaf, 28% to the stem, and 34% to the root. In the same period, the treatment of 250 kg ha^−1^ of P differed from the others, with the highest average concentration of sugar in leaves and roots.

#### 2.3.2. Reducer sugar (RS)

The concentration of reducing sugars (RAs) in leaves, stems, and roots varied between treatments according to each collection period. The highest average concentrations of RS were observed in the leaves, except in the month of October, which occurred in the stem. (Figure 4).

The RSs peak occurred in October, the dry season, where an average of 36% were allocated to leaves, 39% to stem, and 25% to root, between treatments. In February, the rainy season, the distribution of RSs presented different patterns with an average of 62% RSs in the leaves, 3% in the stem, and 35% in the root.

In October, the control and treatment of 50 kg ha^−1^ of P showed increase in RS concentrations in leaves from 47% to 55%, in stems from 99% to 97%, and in roots from 9% to 28%, respectively, compared with the observation of the previous one (August). The treatment with 250 kg ha^−1^ of P o RSs was distributed proportionally throughout the plant (leaf/stem/root). In this period, it is also observed that the foliar concentration had a slight decrease, while those of the stem and root increased by 51% and 79%, respectively, compared to the previous observation. In February, in the rainy season, the concentration of reducing sugar increased in the leaves and decreased drastically in the stem in all treatments.

#### 2.3.3. Sucrose

Sucrose concentrations in leaves, stems, and roots varied between treatments according to each collection period. The highest sucrose concentrations occurred in the month of August and decreased in all parts of the plant as precipitation decreased (Figure 5).

In October, during the dry season, sucrose partitioning in the plant averaged 25% in the leaves, 45% in the stem, and 30% in the root between treatments. In February, the rainy season, the average distribution of the plant was 29% on the leaf, 35% on the stem, and 36% on the root between treatments. In October, the highest concentrations of sucrose in the stem were found in the control and in the roots in those with 250 kg ha^−1^ of P. In the rainy season, we did not observe large variations in sucrose concentrations between treatments.

#### 2.3.4. Starch

Starch concentrations in leaves, stems, and roots varied between treatments according to each collection period. In the leaves, little variation is observed between treatments and evaluation periods. In the stem, there was a reduction in starch concentration in October, with a new increase in February, the rainy season. In the roots, the highest concentrations of starch were recorded in the dry season, August and October (Figure 6).

In October, during the dry season, the starch distribution in the plant varied on average by 19% in the leaves, 24% in the stem, and 57% in the roots between treatments. In February, the rainy season, partitioning averaged 11% in leaves, 52% in stems, and 37% in roots, between treatments. The control and those with 50 kg ha^−1^ of P showed the highest starch concentrations in the last collection period, with the highest concentration in the stem (49%). The treatment with 250 kg ha^−1^ of P showed higher concentrations of starch in the root than the others in all evaluation periods.

## 3. Discussion

### 3.1. Dry Matter Production in Response to Phosphate Fertilization in the Seasons

At the end of the experiment, we registered the highest mean dry matter in leaves and stems was unfertilized plants. In the control, the phosphorus contents in the leaves did not vary between the periods, except in the third evaluation, where we observed a slight drop, while in the roots the contents decreased with the passage of time.

*C. guianensis* normally occurs in tropical forests with highly weathered soils and low P availability. Despite these conditions, some species develop normally due to adaptations for phosphorus (P) acquisition [11,12,13]. These adaptations include the release of inorganic P (Pi) from vacuolar reserves; association with ectomycorrhizal fungi; redistribution of Pi; secretion of organic acids (OAc) and phosphatases to release Pi from soil organic matter, among others [14,15,16]. It is possible that such strategies (a topic not addressed in this study) can be used by *C. guianensis*, which justifies the growth and P levels found in control plants.

Species efficient in the use of P produce a high yield per unit of P absorbed since they have a low internal demand for normal metabolic activities and growth as a strategy. Therefore, these species are capable of producing high yields at relatively low P levels [17]. Also, the organic matter added at the beginning of planting may have made phosphorus available to the plants. Apparently, the control plants absorbed P from organic matter and stored it in the leaves, where the highest levels of the element were found, with little variation between periods. Tolerant plants can secrete organic acids (OAc) and phosphatases, which increase the solubility of inorganic P (Pi) and release it from soil organic matter [14].

Soluble phosphorus is absorbed and can be metabolized or stored in vacuoles—mainly in inorganic form—for use at later stages of plant development and in situations where soil P availability is low [18,19]. We observed this behavior in the present study with *C. guianensis*, considering that in December there was a decrease in P content, mainly in the root, followed by a significant increase in dry matter production in the subsequent collection. Possibly, the availability of P decreased and the plants needed to access the stored P for growth under favorable conditions (rainy season). 

Normally, it is expected that the increased supply of P in the soil will alleviate the limitation of the element and stimulate plant growth [20]. However, in this study, phosphate fertilization does not seem to have benefited the dry matter production of *C. guianensis* plants. Despite having the highest levels of foliar P from the second collection, plants with 250 kg ha^−1^ of P had the lowest average dry matter in leaves and stems. Experimental evidence on tree seedlings from tropical forests shows that phosphate fertilization can increase tissue P content, but not necessarily stimulate plant growth [21,22,23]. Mao et al. [24] reported that long-term P addition had no significant effect on seedling growth of seven understory tree species, although P concentrations in plant tissues were high. 

One hypothesis that explains the low growth of fertilized plants (250 kg ha^−1^) is the limitation of other nutrients caused by the application of P [20]. In the literature there are registers of the phosphorus (P) antagonistic interaction in the soil with zinc (Zn), iron (Fe), copper (Cu), and manganese (Mn); that is, the increase in the Pi content decreases the availability of these elements. for plants [25,26,27,28]. Thus, P fertilization can affect the use of these minerals by plants, causing a nutritional imbalance affecting plant growth.

In the root, the phosphorus content of plants with 250 kg ha^−1^ was higher than that of the other treatments in all evaluated periods, with a peak recorded in August. In the same month, the root had a higher average of the element than the average found in the leaves, suggesting that initially, *C. guianensis* plants tend to accumulate phosphorus in the roots. This peak is followed by a fall in the subsequent observation (October), together with an increase in leaf content, indicating that there was a translocation of the element to the shoot. The production of root dry matter of this treatment increased proportionally with time, with a higher average at the end of the experiment. 

Plants that grow at different concentrations of P develop different adaptive mechanisms. In most experiments with phosphorus, root growth is related to its deficiency in the environment, where the low availability of the element stimulates root growth [14,29,30]. However, phosphorus adequate levels in the soil can also stimulate the growth and development of roots, allowing greater efficiency in the use of water and ionic absorption of other elements [31]. In this study, the highest level of fertilization stimulated the production of root dry matter at the end of the experiment.

At the end of the experiment, the phosphorus content of *C. guianensis* plants was equal between treatments (*p* > 0.05). The addition of phosphate fertilizers resulted in an immediate increase in the P concentration of the soil solution, which boosted the absorption and accumulation of P in the treatment of 250 kg ha^−1^. However, soluble phosphate became insoluble due to adsorption-sorption and/or precipitation processes [32]. The retention and transformation of P in soils are highly dependent on the physicochemical properties of the soils—calcium, aluminum, and iron being the main cations involved in these adsorption reactions [33].

Dry matter production was directly proportional to time in all treatments. There was a moderate increase in the first three collection periods, followed by a significant increase (>100%) in the last evaluation, corresponding to the rainy season. These biomass dynamics are linked to precipitation since plant growth is related to the direct and indirect effects of water restriction [34]. Under low soil water availability, stomatal closure occurs. This reduces carbon assimilation and photosynthesis rates, which consequently affects growth [35,36].

Gonçalves et al. [37], in a study of water stress with *C. guianensis*, found a decrease in photosynthesis and stomatal constancy under a water deficit of 21 days and rapid recovery after rehydration (4–8 days). Oliveira and Marenco [38] observed decreases in photosynthetic rates and biomass allocation in *Carapa surinamensis* under water stress, as well as increased growth under irrigation. Casaroli et al. [34] reported decreased transpiration of African mahogany (*Khaya ivorensis* A. Chev.) during the dry season and increased growth when irrigated. Although the gas exchange was not evaluated, it is suggested that *C. guianensis* plants probably decreased stomatal conductance to avoid water loss. This decreases the photosynthetic rate during the drought and restores gas exchange at the beginning of the rains, stimulating growth.

### 3.2. Total Soluble Sugar Concentrations in Response to Phosphate Fertilization in the Seasons 

In *C. guianensis* plants, there was a tendency for soluble sugars to accumulate in the leaves. The only exception to this was that in October, the dry season, TSS accumulated on the stem. Despite the benefits of phosphorus in the primary metabolism of plants, fertilization had little influence on TSS concentrations in leaves and roots. In the stem, there was no difference between treatments (*p* > 0.05). The greatest variations were observed between the evaluation periods, which were different in all treatments.

The high concentration of TSS in the stem during the second collection, with low precipitation, did not stimulate the growth of *C. guianensis*, regardless of the treatment applied. During stress (water restriction), plants slowed their growth in the short term in favor of storing soluble sugars, as a possible strategy to maintain their metabolic activities and osmoregulation, ensuring their long-term survival [39].

Carbohydrate storage in plants varies between species and regional climates [40]. The conventional theory of storage dynamics is based on the fact that non-structural sugar reserves decrease throughout the dormant/drought season [41]. However, studies carried out with seedlings and trees from tropical forests show the opposite behavior: the reduction/inhibition of growth and an increase in the concentration of non-structural and soluble sugars in the stem under drought, as a survival strategy in adverse conditions [39,42,43,44]; *C. guianensis* plants displayed similar behavior.

The increase in the concentration of soluble sugar in the stem during periods of water deficit is an advantage for plants for the following reasons: it helps in maintaining basic plant functions such as osmoregulation, maintains cellular turgor [8], acts as a signaling substance, allowing adaptation to change environmental issues [45]. Sapes et al. [46] showed that the depletion of non-structural sugars, including soluble sugars, impairs the drought tolerance of seedlings of *Pinus ponderosa* Douglas ex C. Lawson. The authors observed increases in stomatal conductance and the accelerated loss of water associated with the depletion of sugars.

In addition to the benefits of osmotic adjustment, the storage of soluble sugars in the stem allows an accelerated recovery after the return of precipitation. Tomasella et al. [47] reported a positive correlation between post-stress (drought) hydraulic recovery and consumption of soluble sugars. This strategy is adopted by plants to partially restore water transport, increasing stomatal conductance and growth. *C. guianensis* plants showed similar behavior, with a decrease in soluble sugar concentrations and a significant increase in stem dry matter, after the return of the rains in December.

In the roots, there was also an increase in the concentration of TSS in October (dry season), compared to the previous period, and a decrease in the following period. In some cases, plants tend to allocate soluble sugars to roots during water stress to facilitate water uptake and survival but decrease this allocation over time when carbon consumption is greater than supply. Higher concentrations of TSS are beneficial to maintaining stomatal closure and promoting root growth and thus optimize water uptake [8,9,36].

### 3.3. Reducing Sugar Concentrations in Response to Phosphate Fertilization in the Seasons

The highest concentrations of reducing sugars occurred in October, dry season, in all treatments, with emphasis on the control and plants with phosphorus application of 50 kg ha^−1^, which presented high rates, mainly in the stem and leaves. The accumulation of reduced form sugars—such as glucose, sucrose, fructose, and fructans—function as osmoprotectants in stressful conditions such as drought [48,49].

The accumulation of reducing sugars in October, the dry season, may also be related to the reduction in the use of assimilates, since the growth was discreet in this period, considering the low production of dry matter. As water availability decreased, photosynthesis was possibly reduced, inducing hydrolysis of accumulated sucrose. This explains the increased amounts of reducing sugars and reduced amounts of sucrose in the leaves [50]. The fractionation of sucrose into two molecules (glucose and fructose) doubles the number of osmotic molecules active substances in the vacuole. This fact facilitates the uptake of water and the increase in turgor as driving forces for cell expansion [51].

Plants fertilized with 250 kg ha^−1^ P exhibited the lowest concentrations of reducing sugars in leaves and stems, compared to the other treatments, in the October dry season. This is related to overexpression of hexokinase activity under high phosphorus levels in leaves. Phosphorus is used to form ATP, a necessary substrate for hexokinase activity [52]. This enzyme has a catalytic function and provides phosphate hexoses for glycolysis, such as glucose 6-phosphate and fructose 6-phosphate. Thus, hexose sugars (glucose, fructose, and mannose) are converted into their phosphorylated forms, resulting in a low concentration of reducing sugars [52,53,54]. 

In this same evaluation period, the reducing sugars of plants with 250 kg ha^−1^ P tended to be allocated in the roots, which differed from the other treatments, with the highest average concentration in this organ. This increase can be attributed to the energy consumption by the roots of *C. guianensis* to maintain the cytoplasmic osmotic potential of the root cells [55]. In addition to the water deficit, the high concentration of phosphorus in the roots caused an osmotic imbalance. Consequently, there was a stimulation of the production of reducing sugars to assist in the osmotic adjustment and translocation of phosphorus to other parts of the plant.

In the month of February, rainy season, there is a significant increase in foliar reducing sugars in the fertilized plants, compared to the control. This behavior is associated with the normalization of photosynthetic activity, with increased precipitation and benefits of phosphorus on metabolism—factors that stimulate carbohydrate production and help restore growth [6,56,57]. There was a marked reduction in stem concentration in all treatments. Tomasella et al. [47] report that hydraulic recovery is positively correlated with the depletion of sugars during rehydration, as these would be required as a substrate for stem respiration and construction of new xylem tissue.

### 3.4. Sucrose Concentrations in Response to Phosphate Fertilization in the Seasons 

Sucrose concentrations in *C. guianensis* plants tended to decrease with decreasing precipitation, regardless of the treatment used. In October, the dry season, sucrose was observed allocated in the stem.

Sucrose is the main end product of photosynthesis in most plants, being translocated through the phloem from the source tissues (mature leaves) to the draining organs to maintain an adequate level of energy and produce carbon skeletons used in growth. Furthermore, it acts as a signaling molecule and osmolyte to prevent tissue damage under limited water availability [58,59].

Due to its diverse functions, mainly in osmoregulation, sucrose tends to increase in plants under water stress [58,60,61]. However, the results found in this study show a decrease in sucrose with a decrease in precipitation. These reductions can be attributed to a decrease in the photosynthetic rate and in the conversion of sucrose to reducing sugars. As the main product of photosynthesis, sucrose is directly dependent on CO_2_ fixation rates. Thus, water deficit can inhibit photosynthesis and interrupt sucrose supply [62].

The decrease in sucrose in leaves, stems, and roots was accompanied by an increase in reducing sugar concentrations during the month of October. We suggest that *C. guianensis* converts sucrose to reducing sugars to improve osmotic regulation capacity under water stress [49]. Sucrose can break down into other sugars such as glucose, fructose, and trehalose 6-phosphate (reducers) [59]. Under stress conditions such as drought, CO_2_ assimilation and photosynthesis generally decrease, while respiration increases which results in the accumulation of glucose and fructose, used in this process, to the detriment of sucrose [63]. Thus, the decrease in sucrose in *C. guianensis* may be related to the decrease in photosynthetic rate and to the osmotic adjustment of the plant.

### 3.5. Starch Concentrations in Response to Phosphate Fertilization in the Seasons 

In *C. guianensis* leaves, starch concentrations varied little over time. In the stem and roots, they tended to decrease with the decrease in precipitation. Starch was stored in the roots, with lower averages occurring in December, the rainy season, due to two factors: first, starch was metabolized during the dry period. Second, during the rainy season, starch tends to be stored in organs with greater growth activity, such as the stem and leaves.

Starch is the main storage form of carbohydrates in plants, commonly associated with organs such as roots, stems, and seeds. It is stored throughout the seasons to aid growth or seedling establishment early in growth and to provide energy when photosynthesis may potentially be limited [64]. Starch may also play an indirect role as a source of osmotically active organic compounds and energy. Under environmental stress, such as drought, starch is often degraded, resulting in a subsequent increase in soluble sugars (sucrose, glucose, fructose, etc.) in stressed tissues to aid in osmotic adjustment [64,65]. This explains the reduction in starch concentration and increases in AST in the stem of *C. guianensis* in October, suggesting starch partitioning for osmotic adjustment. However, it is unlikely that this breakdown of starch is solely responsible for the increase in soluble sugars, given that in the roots, starch concentrations did not change and AST concentrations increased when compared to the previous observation (August). 

Starch degradation is a common and expected response of plants under stress (drought) and contributes to the accumulation of soluble sugar [61,64]. Contrary to expectations, the starch concentrations in the leaves increased in October. In the roots, there was no difference in relation to the previous observation in August, except for the treatment of 50 kg ha^−1^ P. The pattern of carbohydrate allocation varies between species, in some, starch can accumulate due to decreased demand, as a consequence of growth limitation [64]. 

Deng et al. [66] related the increase in soluble sugar and starch content in *Pinus massoniana* seedlings submitted to drought with growth inhibition, which resulted in the accumulation of starch in all organs. Piper et al. [39] reported, in conifers, similar results of carbohydrate accumulation and reduced growth in dry conditions. The authors suggest a “competition” for carbohydrates between growth and storage, where carbohydrates can be upregulated under drought to maintain turgor and long-distance vascular integrity in the xylem and phloem. This may be a strategy used by *C. guianensis*, considering starch storage and low dry matter production during low rainfall.

Starch maintenance levels also contribute to the rapid recovery of plants after short disturbances in the field environment (variable water availability, stem damage, herbivory, and pathogen infection), as starch acts as a carbon reservoir [67].

## 4. Materials and Methods

### 4.1. Study Area

The experiment was carried out at the Experimental Field Unit (UEC), belonging to the Federal University of Western Pará, located in the Municipality of Santarém, Km 37 of the Santarém Curuá-Una/PA-370 Highway, with coordinates 02°24′52″ S and 54°42′36″ W, from January 2020 to 5 February 2021.

### 4.2. Plant Material Preparation

The fruits of *C. guianensis* came from 20 mother trees from a plantation located at the Experimental Station of Curuá-Una of the Federal University of Western Pará, in the community of Barreirinha, located on the right bank of the Curuá-Una River, Prainha, Pará state. After collection, we sorted the seeds, those with damage or perforations were discarded. Sowing seeds was carried out in plastic trays containing black earth and sand, in a 1:1 proportion, kept in a covered and semi-shaded environment. After 15 days of germination, we transported the seedlings to plastic bags with a substrate composed of a mixture of black earth, rice straw, and sand in a 2:1:1 ratio, where they remained for three months for acclimatization and development. Irrigation took place on alternate days according to the field capacity of the substrate. After this period, we measured the seedlings, and those with a height of 50 cm (±5 cm) were selected for planting.

### 4.3. Experimental Design and Treatments

We designed the experiment in a completely randomized way, in split plots in time. Our plots were submitted to three treatments—three phosphorus dosages (0, 50, 250 kg ha^−1^). The subplots consisted of the evaluation periods (August, October, December, and February). We repeated each treatment (dosage levels) five times, with each plant being an experimental unit, totaling 60 plants. We used simple superphosphate as a source of phosphorus.

### 4.4. Area Preparation and Conduction Experiment

For the fertility assessment, we collected samples from the subsurface layer of soil, from 0 to 20 cm deep. We analyzed chemical attributes: pH in water; phosphorus (P), potassium (K), calcium (Ca), magnesium (Mg), potential acidity (H + Al), effective cation exchange capacity (CEC); the sum of bases (SB), a saturation of bases (V%) and aluminum (m%), following the methodology of RAIJ et al. [68]. As for the granulometry, we determined the contents of sand, silt and clay [69]. The results obtained are shown in Table 1.

Due to the chemical analysis of the soil, limestone with a relative power of total neutralization equal to 90% was used to raise the base saturation of the soil to 50%. The application was individual in each pit and incubated for a period of 40 days, before planting. The pits had dimensions of 50 × 50 × 50 cm, with a spacing of 3 × 3 m.

After three months of acclimatization, the seedlings were planted, with full exposure to irradiance and local climatic conditions. The substrate used for planting consisted of soil from the site itself, with the addition of organic matter (ground açaí seeds), in the proportion of 2:1, and application of phosphate fertilizer. After seven months of planting, we performed the first data collection, in the eighth month, and subsequent evaluations took place at 60-day intervals.

### 4.5. Climate

The region has an Ami climate in the Köppen system, that is, humid tropical with annual temperature variation below 5 °C, with annual rainfall between 1900 and 2400 mm [70]. The region has a rainy season between December and June (higher precipitation) and a dry season (less precipitation) from July to November (Santos et al., 2021), with the occurrence moderate of at least one month with precipitation value. average less than 60 mm [71]. The relative humidity of the air presents values above 80% almost every month of the year [72].

Meteorological data regarding solar radiation, monthly maximum, and minimum temperature were acquired from the National Aeronautics and Space Administration (NASA) Prediction of Worldwide Energy Resource (POWER) database [73]. We obtained the monthly rainfall accumulation on the website of the National Institute of Meteorology (INMET) [74]. Due to seasonal variability, two collection periods were selected from each season for a more detailed analysis of the strategies and use of carbohydrates by *C. guianensis* plants. The collection periods occurred in the months of August and October 2020, corresponding to the dry season, and December 2020 and February 2021, the rainy season (Figure 7).

### 4.6. Evaluated Variables

#### 4.6.1. Dry Matter

In each period, five plant samples were taken from each treatment, which were subdivided into leaves, stems, and roots. Then, all samples were washed with water to remove any adhered particles. We sort the roots to avoid losses, using 2 mm mesh sieves. After washing, the samples were washed again with distilled water and then dried in an oven at 60 °C until they reached constant weight.

#### 4.6.2. Phosphorus Content

The P extract was obtained through nitroperchloric digestion of leaf and root dry matter samples. The levels were quantified spectrophotometrically (UV-Visible), by reading the color intensity of the phosphomolybdic complex [75].

#### 4.6.3. Carbohydrate Concentration

The carbohydrates determination in plants was performed individually in leaves, stems, and roots, from dry matter samples. The extraction of macro and micromolecules was performed using phosphate buffer (0.1 M at pH 7). After obtaining the crude extracts, the pellet was resuspended with potassium acetate buffer (200 mM at pH 4.8) to extract the starch. Sucrose extraction was performed according to the method of Van Handel [76]. The quantification of the concentrations of total soluble sugars (AST) and starch was obtained using the anthrone method [77]. The reducing sugar (RA) was determined by the dinitrosalicylic acid method [78].

#### 4.6.4. Data Analysis

We tested the data for normality by the Kolmogorov-Smirnov test. The results were also submitted for analysis of variance (ANOVA). The means differences between treatments within the collection periods and between months were evaluated by the Scott-Knott test at 5% probability. Statistical analyzes were performed using the computer program SISVAR [79]. We used Sigmaplot 11.0 software to render the plots.

## 5. Conclusions

In the fertilized plants, the tissue phosphorus content of the leaves and roots increased until the month of October and, from this period on, began to decrease. In the roots, the element content decreased over time in all treatments.

*C. guianensis* seedlings were not very responsive to phosphorus addition, after 14 months of the experiment. Control plants (not fertilized) had the highest average dry matter in leaves and stems.

*C. guianensis* seedlings tend to increase the reserves of total soluble sugars and reducing sugars in certain parts of the plan, in periods of low water availability, considering the increase in the concentration of these sugars in the stem and roots of all treatments in the month of October, dry season. On the other hand, sucrose concentrations decreased in all parts of the plant, and starch was allocated to the roots, in the same period.

In December, the beginning of the rainy season, the concentrations of the evaluated carbohydrates decreased in all treatments and increased in the following evaluation in February, with the exception of the stem reducing sugar.

## Figures and Tables

**Figure 1 plants-11-01956-f001:**
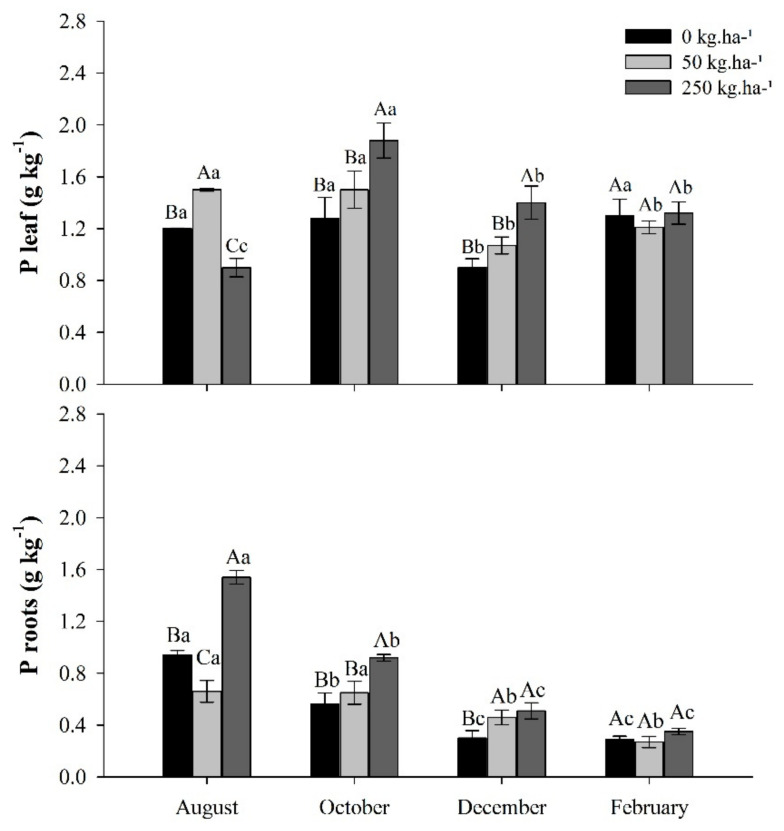
Phosphorus (P) content in leaves and roots of *C. guianensis* plants fertilized with phosphorus at different collection periods. Means followed by distinct letters differ from each other (*p* < 0.05), where uppercase letters represent the differences between treatments and lowercase letters between the evaluation periods.

**Figure 2 plants-11-01956-f002:**
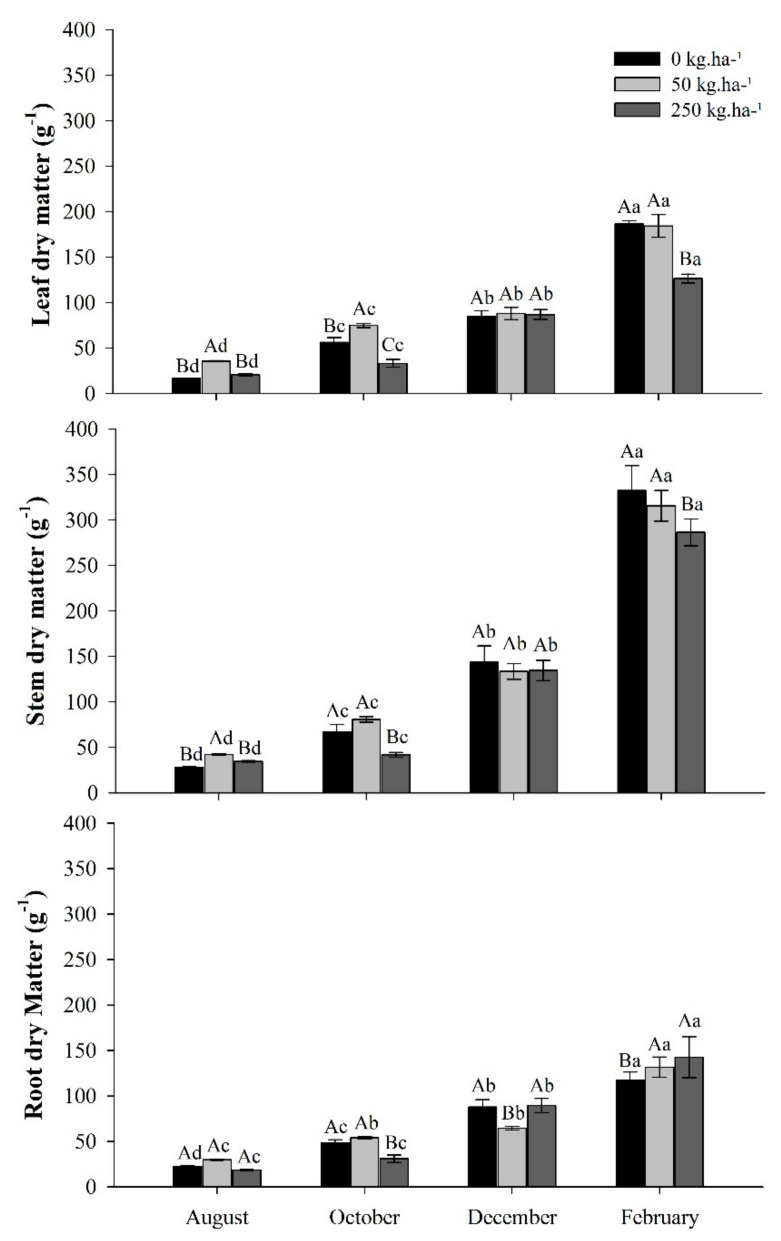
Dry matter of leaves, stem, and roots of *C. guianensis* plants fertilized with phosphorus in different collection periods. Means followed by distinct letters differ from each other (*p* < 0.05), where uppercase letters represent the differences between treatments and lowercase letters between the evaluation periods.

**Figure 3 plants-11-01956-f003:**
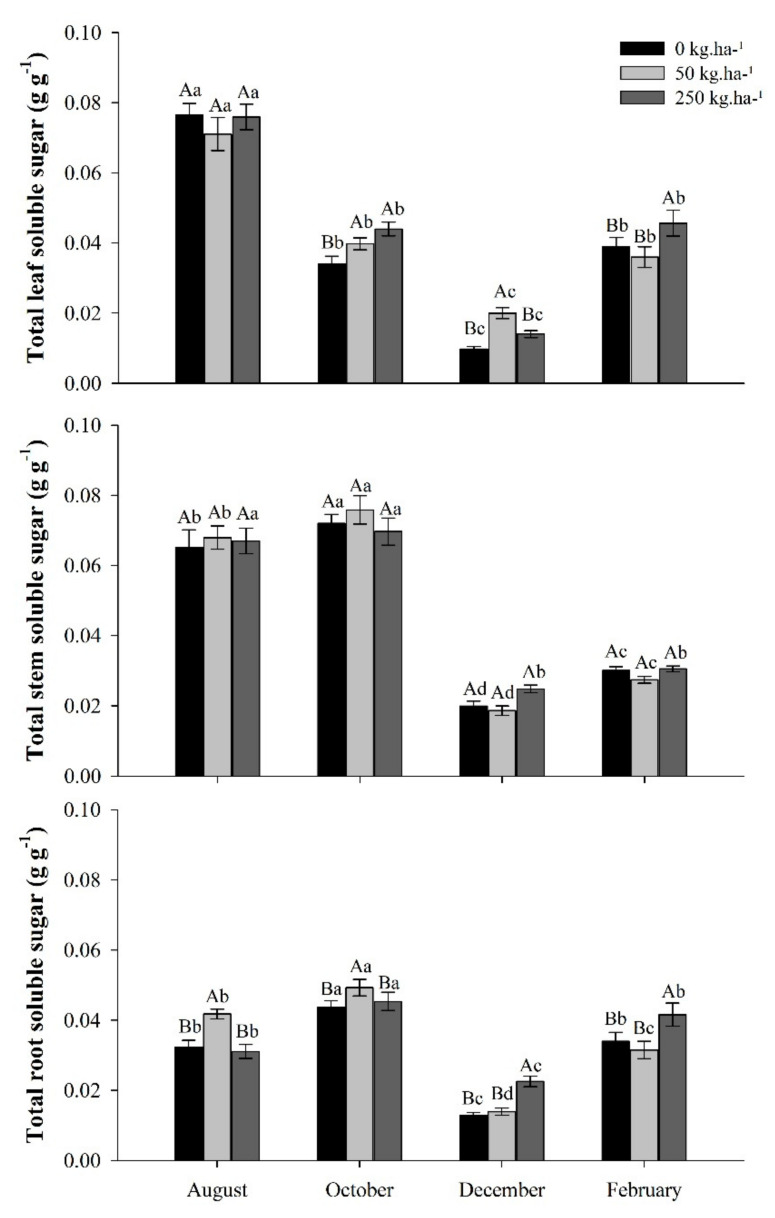
Total soluble sugar from leaves, stems and roots of *C. guianensis* plants fertilized with phosphorus at different collection periods. Means followed by distinct letters differ from each other (*p* < 0.05), where uppercase letters represent the differences between treatments and lowercase letters between the evaluation periods.

**Figure 4 plants-11-01956-f004:**
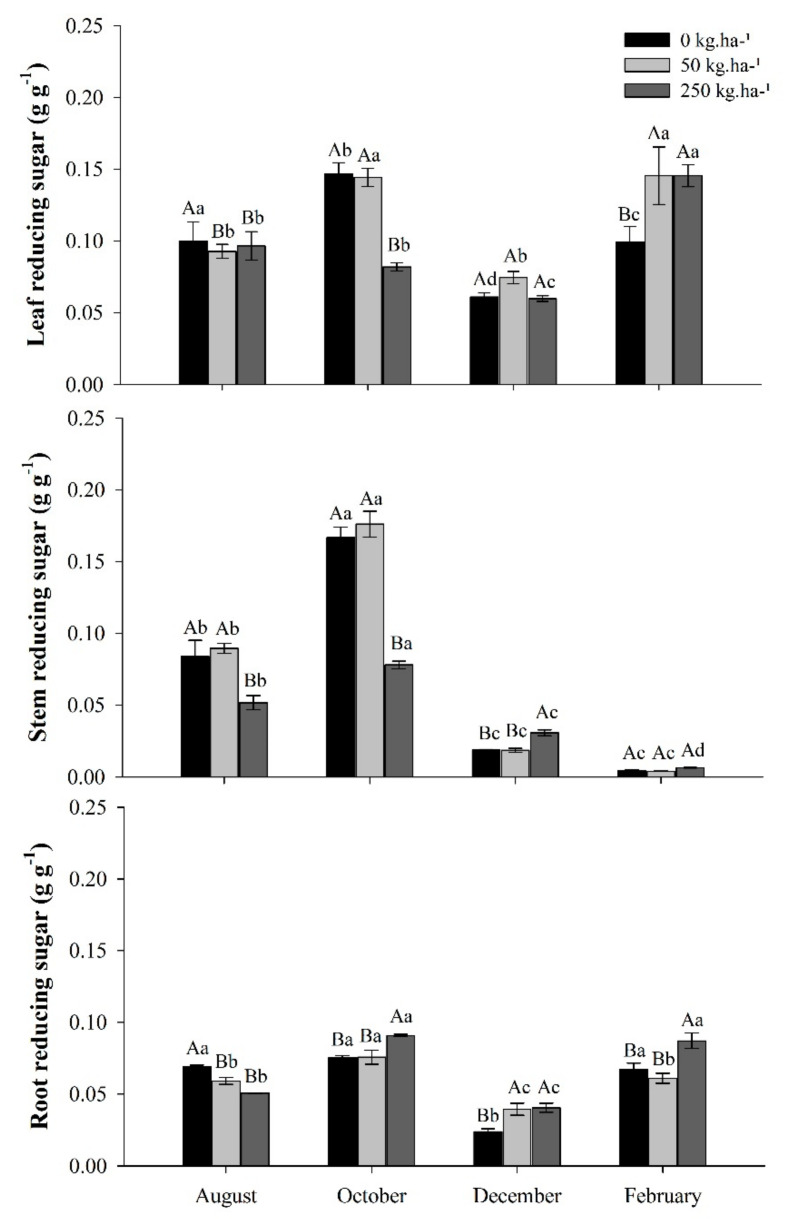
Reducing sugar in leaves, stems, and roots of *C. guianensis* plants fertilized with phosphorus at different collection periods. Means followed by distinct letters differ from each other (*p* < 0.05), where uppercase letters represent the differences between treatments and lowercase letters between the evaluation periods.

**Figure 5 plants-11-01956-f005:**
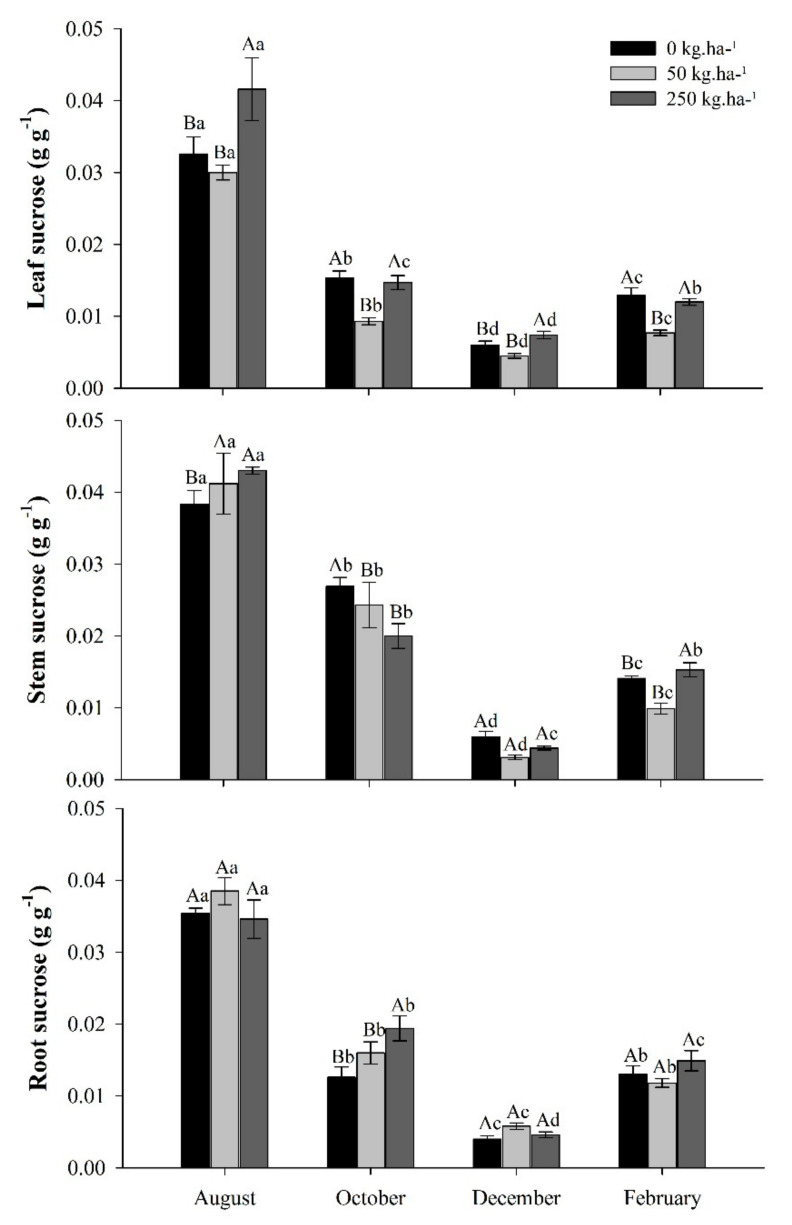
Sucrose from leaves, stems, and roots of *C. guianensis* plants fertilized with phosphorus at different collection periods. Means followed by distinct letters differ from each other (*p* < 0.05), where uppercase letters represent the differences between treatments and lowercase letters between the evaluation periods.

**Figure 6 plants-11-01956-f006:**
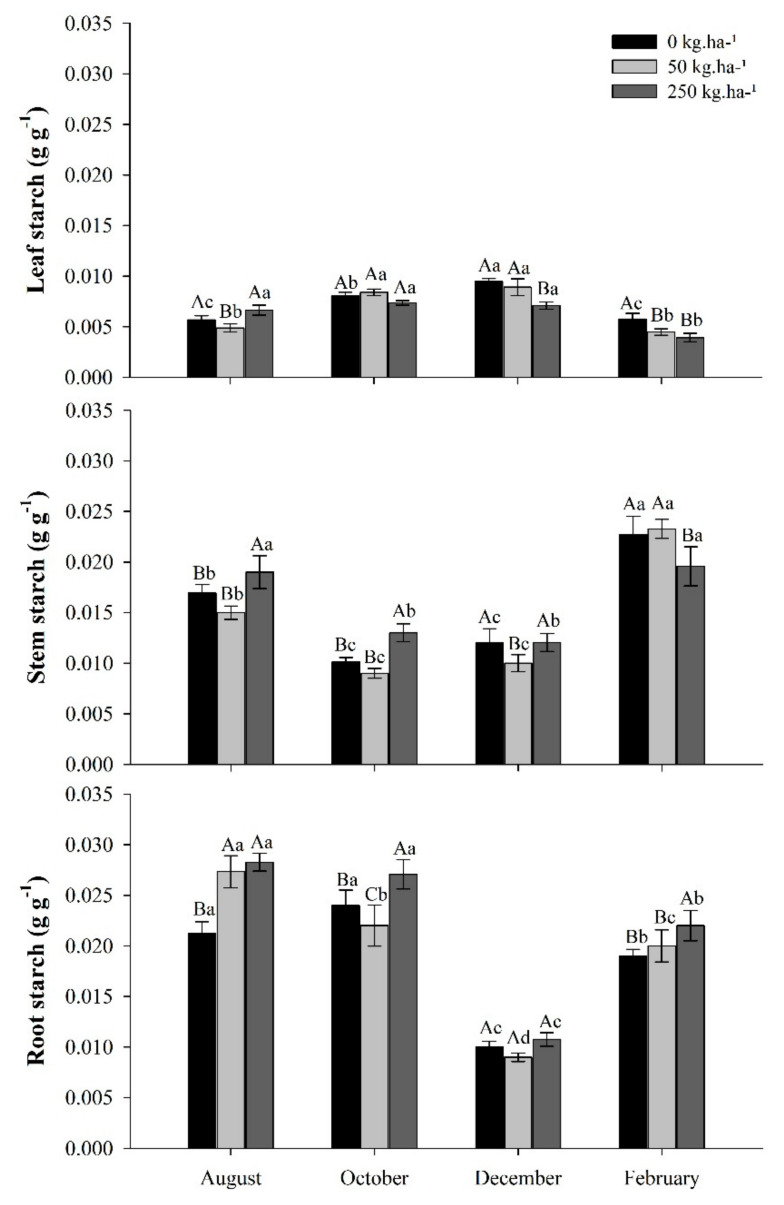
Starch in leaves, stem, and roots of *C. guianensis* plants fertilized with phosphorus, at different collection periods. Means followed by distinct letters differ from each other (*p* < 0.05), where uppercase letters represent the differences between treatments and lowercase letters between the evaluation periods.

**Figure 7 plants-11-01956-f007:**
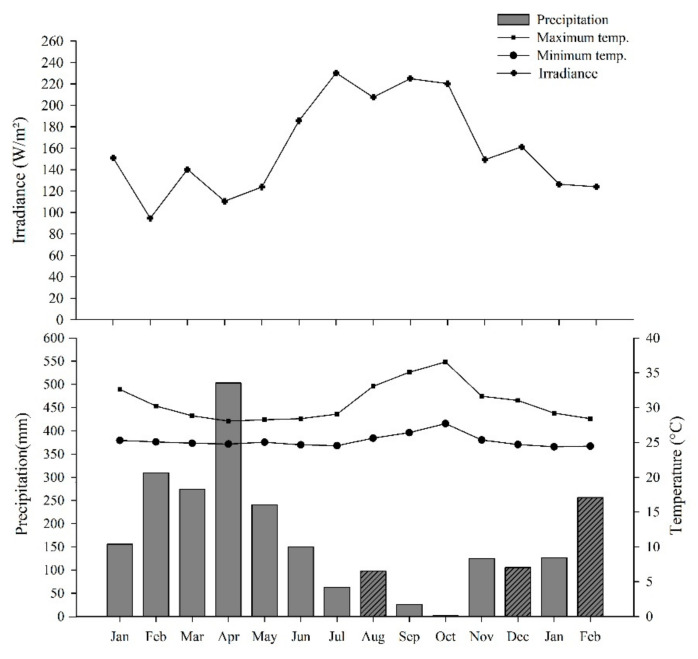
Precipitation (mm), maximum and minimum temperature (°C), and monthly solar irradiance (W/m²) during the study period, from January 2020 to February 2021.

**Table 1 plants-11-01956-t001:** Chemical attributes and soil granulometry in the 0–20 cm depth.

Chemical Analysis										
MO	P	K	Ca	Mg	H + Al	CTC (t)	SB	m	V	pH
g kg^−1^	mg dm^−3^		cmol_c_ dm^−3^				%			H_2_O
0.9	6	28	0.5	0.3	5.1	1.7	0.9	48.3	14.8	4.3
Granulometry										
Total sand				Silt			Clay			
g kg^−1^										
278				146			576			

## Data Availability

The data presented in this study are available on request from the corresponding author.

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
