# Peer review of "Seasonality and Phosphate Fertilization in Carbohydrates Storage: Carapa guianensis Aubl. Seedlings Responses"

_plants, 2022, doi:10.3390/plants11151956_

Round 1
Reviewer 1 Report
The authors report on the effects of two phosphate fertilization levels on plant properties. The also say they examine how the results change for plants collected during different months (August, October, December and February). The response variables are P content in leaves, P content in roots, leaf, stem, and root dry mass, total soluble sugars, reduced sugar content, sucrose, and starch. The work is important but the specific question/hypothesis is not clear.
Generally the writing is clear. There are few instances of grammatical issues such as line 18 (we should be upper case) and line 489 (it is suggested the development of new experiments..).
On line 20, the abstract reads that tissue prosphorus content increased according to fertilization levels but looking at Figure 1 the statement is only accurate for October and December for leaf and October - February for the roots. This is a general issue in the manuscript - the precise stated results in several instances do not agree with the data presented in the figures.
Another example is the statement on line 21: The control had higher dry matter production. This is only true for some seasons and some parts of the plant (for example the opposite trend is seen for the roots). If the authors are referring to the total masses, they should provide stats and averages for the total plant dry mass somewhere in the MS.
Yes another example can be found on line 485 (in conclusions). The authors need to be more specific/precise when stating conclusions and offer the appropriate statistics to support them. And if the trends are only observed for some parts of the plant and not for all seasons, the authors should exercise caution when drawing conclusions. The missing error bars further complicate the analysis.
The introduction provides the necessary background for the study and a rationale for studying carbohydrate storage in Carapa guianensis.
The introduction states an objective: ...to analyze the production and allocation carbohydrates in ... submitted to different doses of P, during the rainy and dry seasons). The introduction lacks a clear question and/or hypothesis.
The results starts with stats for an interaction between P and evaluation period but the meaning or purpose of stating this interaction are not clear.
The second sentence in results.. There were influences on dry matter production.. is not clear because it is not clear what was influencing these parameters.
The figures (1- 6) are missing error bars. This makes me wonder whether the data is robust enough to draw the conclusions presented.
The discussion is thoughtful and brings in appropriate literature. My concern about the validity of the conclusions tempers my enthusiasm - as described above.
On line 225 there is a statement comparing root P content to leaves but the statistics reported in Figure 1 do not report on differences between root vs. leaf concentrations.
In a similar vein, the authors make comparisons between seasons but the stats shown in figures are post hoc stats and only allow for comparisons between specific collection dates/treatments. When making general conclusions about differences in response variable values between seasons, the authors should provide the appropriate average values and statistics.
It is interesting that the P fertilization did not result in bigger plants and the authors offer a robust discussion with possible reasons for why this might have been.
In the methods, some clarifications are needed. Sampling the first 20 cm of soil is not clear (maybe language like from the surface is needed). The abbreviations for chemical attributes are needed. Also, abbreviations need to be defined in table 1 (for example MO) and the units cmoc dm3 are likely incorrect. When units of % are shown, the aythors need to specify what they mean.
More importantly, in the methods, it is not clear how long the plants from different collection months grew. On line 438, it states that the seedlings were planted at the beginning of rainy season and planting was evaluated after 7 months. But if the collection points were different (August vs. October etc.) then for all plants to grow 7 months, it seems the planting dates would also need to be different to keep the plant life time and treatment time constant. This should be clarified.
Author Response
Firstly, we would like to thank you for all the considerations and review of the manuscript, which were crucial to improving the text. Requested changes are highlighted in blue in the text.
I want to mention some critical issues. Our objective was to analyze the allocation of carbohydrates to the detriment of phosphate fertilization and the different seasonal periods. The interaction between fertilization and seasonal periods was not evaluated. We evaluated carbohydrates x fertilization and carbohydrates x periods. In view of your questioning, in the item “data analysis”, in the material and methods, we better described performing a test of means between treatments and between periods, separately.
Another point mentioned was the general comparisons between the stations. The purpose of the study was the evaluation the collection periods in each station. Thus, in the parts of the text in which this general comparison took place, the periods (months) were mentioned, specifically, in which a certain behavior occurred, citing which season (dry or rainy) it belonged.
In materials and methods, we better exposed the experiment implementation, that the seedlings were planted three months after germination and that the collections started after seven months of planting with a time interval between and every 60 days. The collection dates were selected based on the climatic seasons of the Western Region of Pará.
Reviewer 2 Report
The manuscript provides significant information about the responses of the P content, dry matter production and allocation of carbohydrates in plants of Carapa guianensis subjected to various does of phosphate fertilization, during the rainy and dry seasons. Considering the Climate and environmental impacts, the outcome of this study helps to introduce sustainable cultivation techniques for Carapa plants which provide high-quality wood as well as oil extracted from their phamaceutically-owned seeds. The whole manuscript is very well written and presented to the reader. Introduction is based on recent findings and “Materials and Methods” are described with sufficient details. The results have been appropriately verified by statistical analysis. Tables and figures are very clear to the reader and understandable. Main results and ideas are well documented, justified and supported by relative references. Minor changes are recommended for section “Materials and Methods” regarding the soil analysis methodology. Based on the above the manuscript can be considered for publication in this journal considering the minor changes.

Author Response
We would like to thank you for all the considerations and review of the manuscript, which were crucial to improving the text. Requested changes are highlighted in blue in the text.